

# An intercalibrated dataset of Total Column Water Vapour and Wet Tropospheric Correction based on MWR on board ERS-1, ERS-2 and Envisat

Ralf Bennartz[1,2], Heidrun Höschen[3], Bruno Picard[4], Marc Schröder[3], Martin Stengel[3], Oliver Sus[3], Bojan Bojkov[5,*], Stefano Casadio[5], Hannes Diedrich[6], Salomon Eliasson[7], Frank Fell[8], Jürgen Fischer[6], Rainer Hollmann[3], Rene Preusker[6], Ulrika Willen[7]

1: Earth & Env. Science Department, Vanderbilt University, Nashville, TN, USA
    2: Space Science and Engineering Center, University of Wisconsin- Madison, USA
    3: German Meteorological Service (DWD), Offenbach, Germany
    4: Collecte Localisation Satellite (CLS), Toulouse, France
    5: European Space Agency (ESA-ESRIN), Frascati, Italy
*: Now at EUMETSAT, Darmstadt, Germany
    6: Institute for Space Sciences, Freie Universität Berlin, Germany
    7: Swedish Meteorological and Hydrological Institute (SMHI), Norrköping, Sweden
    8: Informus GmbH, Berlin, Germany

*Correspondence to*: Ralf Bennartz (ralf.bennartz@vanderbilt.edu)

**Abstract.**

The Microwave Radiometers (MWR) on-board ERS-1, ERS-2, and Envisat provide a continuous time series of brightness temperature observations between 1991 and 2012. Here we report on a new Total Column Water Vapour (TCWV) and Wet Tropospheric Correction (WTC) dataset that builds on this time series. We use a one-dimensional variational approach to
derive TCWV from MWR observations and ERA-Interim background information. A particular focus of this study lies on the intercalibration of the three different instruments, which is performed using constraints on liquid water path (LWP) and TCWV. Comparing our MWR-derived time series of TCWV against TCWV derived from Global Navigation Satellite System (GNSS) we find that the MWR-derived TCWV time series is stable over time. However, observations potentially affected by precipitation show a degraded performance compared to precipitation-free observations in terms of the accuracy
of retrieved TCWV. An analysis of WTC shows further that the retrieved WTC is superior to purely model-derived WTC for all satellites and for the entire time series. Even compared to operational WTC retrievals, which incorporate additional observational data, the here-described dataset shows improvements in particular for the mid-latitudes and for the two earlier satellites ERS-1 and ERS-2. The dataset is publicly available under http://dx.doi.org/10.5676/DWD_EMIR/V001.





## 1. Introduction

ESA's altimetry missions are at the heart of significant progress on oceanography. The combined coverage of high-quality observations by ERS-1, ERS-2, and Envisat spans over more than 20 years from 1991 to 2012. During this period, improvements in instrument data processing as well as orbit and geophysical corrections allowed reaching an

accuracy/sensitivity of 1 cm on instantaneous sea surface height (SSH) measurements and demonstrated the capability to observe a 3 mm/year sea level rise (Ablain et al., 2009).

A major source of uncertainty for radar altimetry is the wet tropospheric correction (WTC). The spatial and temporal variability of water vapour is such that an instantaneous estimation of its impact is needed. To provide the observations required for the WTC is the primary role of the nadir looking Microwave Radiometer (MWR) embedded into the altimetry

missions on board ERS-1, ERS-2, and Envisat. In this context, requirements on accuracy, sensitivity, and long term stability of the atmospheric water vapour observations are particularly strong since altimetry missions require a precision better than 1 cm in WTC (RMS) (Eymard et al., 2005) and a temporal stability better than 1mm/year (Ablain et al., 2009). Note that a total column water vapour (TCWV) contribution of 1 kg/m$^2$ is equivalent to a WTC of about 6.4 mm.

Water vapour also is a highly important climate variable in its own right. The atmospheric water vapour feedback is believed

to be the strongest feedback mechanism in climate change, approximately doubling the direct warming impact of increased CO2 forcing (Cess et al., 1990;Forster et al., 2007). Various groups have reported trends in the amount of columnar water vapour. In particular, over the oceans, a strong trend in TCWV has been observed (Trenberth et al., 2005). TCWV also appears to be a key factor regulating tropical precipitation (Bretherton et al., 2004).

The importance of water vapour in the climate system is recognized by the Global Energy and Water Cycle Experiment

(GEWEX), which is currently performing an assessment of long-term water vapour products  (the GEWEX Water Vapour Assessment G-VAP). Of particular importance for the current study is the recent publication by Schroeder et al. (2016), who provide an overview on existing TCWV datasets and assesses their long-term stability as well as issues caused for example by changes in the observation systems.

The MWR instruments on-board ERS-1, ERS-2, and Envisat are based on very similar architectures and have measured

water vapour over the ocean between 1991 and 2012 and therefore provide a 20+ years dataset of water vapour observations. Data continuity into the future is ensured with ESA's Sentinel-3 series of satellites, the first of which has been launched in February 2016. MWR's two channels located at 23.8 GHz and 36.5 GHz allow for the simultaneous retrieval of TCWV and cloud liquid water path (LWP) as outlined later in this publication. Through the REAPER (REprocessing of Altimeter Products for ERS) project (Gilbert, 2014) [1], ESA has provided significant efforts to produce an up-to-date data record of

well-calibrated MWR observations.

The efforts described herein build on the REAPER dataset and address three inter-connected issues. Firstly, the homogenization and inter-calibration of the MWR data record is studied and an improved inter-calibration is developed.

---

[1] https://earth.esa.int/web/sppa/activities/multi-sensors-timeseries/reaper/



Secondly, using state-of-the-art one-dimensional variational retrievals, a TCWV data product is developed and made available to the community. A revised WTC product accompanies this dataset also. Thirdly, the dataset is validated both against GNSS observations of TCWV as well as in terms of it's meso-scale stability with respect to WTC. The so-derived dataset is made available to the community.

5   This publication is organized as follows: In Section 2 we describe the MWR brightness temperature time series as well as the methods used for retrieving TCWV and WTC. Section 3 addresses the central issue of intercalibration and Section 4 summarizes the validation results. The sensitivity of the retrieval with respect to the background (a-priori) temperature and water vapour profiles is discussed in the Appendix.

## 2. Datasets and Methods

### 2.1. The MWR dataset

The Envisat MWR brightness temperatures have been generated by CLS in 2014 in the framework of the Envisat MWR L1B Expert Support Laboratory (ESL) activities funded by ESA. It consists of a corrected dataset that removes the anomaly that has been observed in version 2.1. The ERS-1 and ERS-2 MWR brightness temperatures used herein are based on the REAPER project (Gilbert, 2014) but have been entirely reprocessed in the frame of the EMiR (ERS/Envisat MWR Recalibration and Water Vapour Thematic Data Record Generation) project. The so-called "first run" REAPER L1B data have been the basis for this reprocessing. Land measurements are discarded and no specific processing is applied in coastal areas so that contamination from land may occur above coastal waters at distances of less than ca. 50 km from land. Such potentially land-contaminated pixels are excluded from the analysis presented here.

### 2.2. GNSS dataset

MWR estimates of TCWV are evaluated against a 2-hourly data set of TCWV measured by ground-based GPS (Wang et al., 2007), composed of the International Global Navigation Satellite System (GNSS) Service (IGS), SuomiNet and GPS Earth Observation Network (GEONET), and hosted by NCEP/NCAR. While the focus of SuomiNet (see e.g. (Ware et al., 2000)) is the US and Central America and GEONET (see e.g. (Shoji, 2009)) is based on Japanese stations, IGS (see e.g. (Byun and Bar-Sever, 2009)) includes about 500 stations distributed globally[2]. The methodology of how TCWV is derived from the measured GPS zenith path delay is described in detail in (Wang et al., 2007). They also found the individual errors to be less than 1 mm, total errors less than 1.44 mm. The GNSS data base version 721.1 includes data from 1995 to 2014. In total, 997 stations are specified. However, not all of them cover the full period. Every station is characterized by latitude, longitude and altitude and provides 2-hourly data that contain day and time (UTC), surface pressure (hPa), atmospheric weighted-mean temperature (K), TCWV as well as information about zenith delay.

---

[2] www.igs.org/network





### 2.3. Retrieval methods

### 2.3.1. TCWV

Combined TCWV and LWP retrievals were performed here are based on a one-dimensional variational (1D-VAR) scheme initially developed at ECMWF by (Phalippou, 1996) with a focus on microwave observations from SSMIS and AMSU. It

was extended by (Deblonde and English, 2003) towards a stand-alone scheme applicable to SSM/I, SSMIS, and AMSU. This scheme was also used in the ESA DUE GlobVapour project (http://www.globvapour.info). Here, its has been modified to derive TCWV from brightness temperatures specifically from the MWR sensor family on-board ERS-1/2 and Envisat over the ice-free ocean. The scheme follows optimal estimation theory considering the uncertainties in the required meteorological background information, forward modelling (radiative transfer simulations), and satellite observations.

The 1D-VAR scheme employed here uses daily global ERA-Interim TCWV and atmospheric temperature and cloud water content profiles and various surface fields as a-priori (background) and first guess information. Information on the choices of various input parameters is found in the next sub-sections.

For the particular case of the MWR, the observation vector consists of the two brightness temperatures at 23 and 36 GHz. The 1D-VAR uses a 74-element state vector including 43 temperature levels, 26 moisture levels, surface air temperature,

surface specific humidity, sea surface temperature, wind speed, and cloud liquid water path. While the inter-dependencies especially of the vertical levels are constrained by the background error covariance, the retrieval obviously remains heavily under-constrained, so that ultimately only TCWV and LWP are constrained by the two observations at 23 and 36 GHz.

As pointed out above, the main information content of the 23 and 36 GHz channels lies in TCWV and LWP, respectively. Given just these two observations, care must be taken not to overfit the solution by optimizing parameters only weakly

related to the two observables (say, the atmospheric temperature profile). While the Jacobian matrix for such less-constrained parameters is in any case small, a cleaner way of eliminating this issue is to set the Jacobian to zero for all parameters that are not desired to be retrieved. This was done for this study for the atmospheric temperature profile and all surface parameters. Therefore only the water vapour profile and cloud liquid water are actively adjusted during the 1DVAR process, whereas all other parameters were treated as fixed background parameters, set to the values prescribed by ERA-

Interim.

The cost function J(x) is defined as:

$$J(\mathbf{x}) = \frac{1}{2}(\mathbf{x} - \mathbf{x}_b)^T \mathbf{S}_b^{-1}(\mathbf{x} - \mathbf{x}_b) + \frac{1}{2}(\mathbf{H}(\mathbf{x}) - \mathbf{y})^T \mathbf{S}_o^{-1}(\mathbf{H}(\mathbf{x}) - \mathbf{y}) \qquad (1)$$

where the first part on the right-hand side determines the cost of the solution with respect to the background and the second

part determines the cost with respect to the observations. For any given retrieval the expectation value of J(**x**) is equal to the number of observations. In the above formulation **y** is the observation vector, **x** the state vector, **x_b** the background state,





**H(x)** the forward model, $\mathbf{S_b}$ the background error covariance matrix, and $\mathbf{S_o}$ the observation error covariance matrix. For further details, see (Rodgers, 2000).

### 2.3.2 WTC

The derivation of the wet tropospheric delay (WTC) follows the analytical procedure laid out in Appendix A. The value of

the wet tropospheric delay depends on the MWR-derived TCWV and on the value Tm (Eq. (15)), which in turn depends on the ERA-Interim temperature profile and the MWR-derived TCWV.

### 2.4 Generation of gridded data

The individual retrievals discussed above were used to calculate global fields of monthly averages of total column water vapour, liquid water path, and brightness temperatures at 23 and 36 GHz on a 2x2° as well as a 3x3° latitude-longitude grid.

Before calculating the monthly means, some filters were applied. A retrieved data pixel was used if a positive total column water vapour retrieval (TCWV>0) was available, meeting two additional conditions: (1) liquid water path larger than -1 kg/m², and (2) a cost function value lower than 5. The last condition effectively removes heavily precipitation-contaminated pixels as well as observations with remaining sea ice or land contribution. Data of at least 20 days were required within a grid cell for monthly mean values to be reported.

In a first step, the grid was set up according to the considered spatial resolution (2°x2° or 3°x3° lat/lon). For each MWR footprint, the corresponding indices of the global fields were calculated from the latitude/longitude information. If all conditions were met (see above), the retrieved and auxiliary values were added to the corresponding grid box. The daily averages were then calculated as the arithmetic mean of all observations within that grid box within one day. If, for a given month and grid-box, more than 20 days had valid observations, the arithmetic mean of those was assigned to be the monthly

mean value. The so-derived monthly mean fields, along with the individual retrievals, are part of the published dataset. All analysis reported within this publication was however performed on individual retrievals.

### 3. Intercalibration and bias-correction

Retrieval of geophysical variables using physical models and optimal estimation procedures requires the elements of the observation vector to be on average unbiased compared to the forward model applied to the true state of the atmosphere.

Comparing first guess simulations with observations, biases include contributions from the following error sources:

- Representativeness of the state vector in the first guess (e.g. representation of clouds in the model),
- Spatial and temporal colocation errors between first guess and MWR observations,
- Calibration biases/errors of the different MWRs,
- Systematic errors and uncertainties in the surface emissivity model.





- Systematic errors and uncertainties in spectroscopy of liquid water absorption, dry air absorption, and water vapour absorption,

- Impact of precipitation contamination and precipitation-ice scattering not accounted for in forward model,

While the first two items on this list have a significant impact on the values reported here, they only play a secondary role for

the retrieval accuracy. As pointed out in Appendix - B, the retrieval is sufficiently independent from the first guess to allow for valid retrievals even if the first guess and prior are relatively far away from the true state of the atmosphere. Furthermore, the relaxed background error covariance criteria formulated also in Appendix -B allow for the retrieval to converge to values corresponding to the observed brightness temperatures.

The latter four items in the above list, while having smaller contributions to the overall bias, are of crucial importance to the

accuracy and long-term stability of a climate data record. These are addressed in an empirical bias-correction scheme as outlined below.

### 3.1. Method

Because of the low sensitivity of the retrieval to the first guess as well as to the background state (see Appendix B), the bias correction proposed here relies on two main assumptions:

- The globally averaged TCWV from ERA-Interim is considered reasonably accurate in terms of its absolute value to provide a reference against which to gauge the average brightness temperature biases of the MWR time series. Note, that we do not make any claims about the long-term stability of the ERA-Interim TCWV or any trends and discontinuities of the dataset. The only assumption we make is that ERA-Interim TCWV on a globally and monthly averaged basis is accurate to within ±2 kg/m$^2$. This assumption is justified from intercomparison efforts such as

Schröder et al. (2013).

- The second assumption is that histograms of instantaneous retrieved LWP must show a significant fraction of negative retrieved LWP, which corresponds to measurement noise around zero LWP for cloud-free situations. For typical bias-free optimal estimation retrievals LWP for cloud-free cases is centred around zero g/m2 with a standard deviation of about 30 g/m$^2$ (see for example (Greenwald, 2009;Bennartz et al., 2010)).

These two constraints can be used to find an optimal bias correction for both channels of each instrument in the following way:

1. For each instrument and month a certain amount of observations out of all observations available were sub-selected. We chose 4% of the total number of observations.

2. For these 4 % we ran a series of retrievals with different biases at 23 GHz and 36 GHz. We ran retrievals for bias

values running from -8 K to 0 K in steps of 1 K for both channels, so that in total 9 x 9 = 81 retrievals were performed on the set of 4% of each data per month.





3. We then found the best choice of biases for 23 and 36 GHz, i.e. the combination of biases for which the difference between background TCWV (ERA-Interim) and retrieved TCWV is smallest and the LWP shows the Gaussian behaviour around zero. This will be the optimal bias value for this particular month.

4. Step 3. was repeated for all months and satellites to find monthly optimal average bias values.

5 Figure 1 and Figure 2 highlight some key methodological issues related to the method. In particular, Figure 1 shows how the histogram of LWP shifts as the bias at 36 GHz varies. It also shows the histograms fitted to the retrieved LWP histograms. This fit was performed on the part of the histogram left of its peak, assuming that all values left of the peak correspond to cloud-free scenes. The super-Gaussian distribution to the right of the peak corresponds to actual clouds. In the particular case shown in Figure 1 the best bias value for 36 GHz would be very close to -6 K, thereby centring the histogram on zero as 10 outlined above.

Figure 2 shows isoplots of TCWV and LWP histogram biases for a full set of monthly retrievals and all combinations of 23 GHz and 36 GHz biases. The optimal set of biases can now be inferred from this histogram as the intersect between the zero TCWV bias isoline (thick, solid) and the zero LWP histogram bias line (thick, dashed) and is located near (-4 K, -7K). The same analysis was performed for all months and instruments. The results are shown in Figure 3 and discussed in the next 15 section.

From Figure 2 it is also noteworthy that the sensitivity of TCWV to biases in 23 GHz brightness temperatures is roughly 1 kg/m$^2$ per 1 K bias. The sensitivity of LWP to biases at 36 GHz is roughly 25 g/m$^2$ per 1 K bias. Note that both variables also exhibit sensitivity to the other frequency although the sensitivity is somewhat smaller as expected.

--- Figure 1 **here** ---

20 --- Figure 2 **here** ---

### 3.2. Bias analysis

Figure 3 shows the outcome of the above-described bias analysis for all instruments and channels. Mean bias values as well as slopes are also listed in Table 1. A negative bias correction value means that the observations are warmer than the simulations, thus the observations need to be corrected downwards. We found the following:

25 • Biases correction values at 23 GHz are between about -4 K for ERS-1 and -2 K for ERS-2 with Envisat being in between these two values.

• ERS-2 23 GHz brightness temperatures show a significant decrease in bias exactly at the time of the instrument's gain drop (indicated by the blue arrow in Figure 3) and a downward trend in bias (slope) for the period afterwards. This strong drop prompted us to separate ERS-2 in a pre- and post-gain time period for which performed a separate 30 analysis (listed in Table 1).

• Envisat shows a slight upward slope possibly over the first half of its lifetime or over its entire lifetime.

• Brightness temperature biases at 36 GHz are comparably stable over time for both ERS-1 and ERS-2, i.e. the regression slopes are very small.



- Envisat shows a strong annual cycle in bias at 36 GHz, which diminishes somewhat after 2008. It also shows a similar trend as it does for 23 GHz. There was no explanation for this behaviour at the time of writing.

The bias values given here are based on optimal comparison between retrieved versus background TCWV as well as constraints made on the histogram of retrieved LWP. While these constraints are physically reasonable and justifiable on
average, it is not advisable to perform monthly bias corrections based on the individual values derived for that particular month. This would by example of TCWV likely result in an over fitting to the ERA Interim time series.

--- Figure 3 here ---

--- Table 1 here ---

However, different biases between two instruments observed in their overlap periods can be clearly attributed to the
instrument calibration, as the background is identical. Similarly the mean bias values reported in Table 1 provide for a reasonable way of addressing the principal error sources outlined above. A first order bias correction is therefore performed subtracting the bias values in Table 1 from the observations.

We note here that biases in the order of -2 K to -5 K (simulations too warm compared to observations) are also reported by ECMWF for monitoring of AMSU-A against their operational forecasting system[3]. It is therefore likely that ERS-2 is
calibrated to within the absolute calibration accuracy of 3 K stated for the instrument. On the other hand, ERS-1 and Envisat both show much larger biases, which might be indicative for remaining calibration issues.

As pointed out above, no assumptions about the long-term stability of ERA Interim should be made in this analysis. The regressions listed in Table 1 therefore cannot conclusively be interpreted as either being caused by natural variability in TCWV or as being caused by instrument drifts.
While a conclusive statement of the origins of the regression slopes and related trends cannot be made, it is interesting to relate the slopes back to the aforementioned retrieval sensitivities. Ignoring the relatively short ERS-2 period before the gain drop, the regression slopes found in Table 1 show values between -0.12 K/yr and +0.1 K/yr. Factoring in the sensitivities of the retrieval the retrieved TCWV, trends observed in global TCWV between the constant bias correction and the regression are expected to be different by roughly -1.2 kg/m$^2$/decade to + 1.0 kg/m$^2$/decade. These numbers provide bounds on how
large observed TCWV trends have to be in order to be considered real, if the current dataset is being used.

## 4. Validation

### 4.1. TCWV

MWR-retrieved TCWV was validated against TCWV derived from coastal GNSS stations. Validation was performed in terms of biases and root-mean-square errors (RMSE) as well as in terms of long-term stability of the dataset. . A total of
30,712 valid collocations were found. The temporal collocation difference was chosen to be at maximum one hour. Note that our dataset excludes observations close to land, so that the distance between a coastal GNSS station and the nearest

---

[3] http://www.ecmwf.int/en/forecasts/charts/obstat/?facets=Parameter,All%20sky%20radiances





MWR observation is somewhere around 100 km. The maximum collocation distance was chosen to be 150 km. Figure 4 shows the coastal GNSS stations used. Figure 5 summarizes the validation results For the entire dataset, we find a relatively small bias of 0.63 kg/m$^2$ and an RMSE of 4.68 kg/m$^2$. If the comparisons were restricted to exclude MWR-retrievals with high LWP (> 200 g/m$^2$) and GNSS stations with altitudes higher than 50 m above MSL were also excluded, the RMSE was

5 reduced to 3.95 kg/m$^2$. LWP-values larger than about 200 g/m$^2$ likely represent precipitating clouds (e.g. Wentz and Spencer (1998)). We therefore reason that remaining issues with precipitation contamination might deteriorate the TCWV retrievals.

--- Figure 4here ---

10 The lower panels in Figure 5 provide time series of average relative differences between MWR and GNSS. Comparing time-series of MWR-GNSS, the expected stability (i.e. the slope of regression line) should be zero. The observed long-term stabilities are +0.68%/decade and  -0.38% for the entire dataset and for the reduced LWP<200 dataset, respectively. Both stability values are not statistical significantly different from zero (P-value for two-sided t-test 0.148 and 0.572, respectively), so that at least for the comparison against GNSS the dataset can be considered stable over time.

--- Figure 5 here ---

Several other tests including separation of daytime from night-time observations did not yield any additional insights into the uncertainties.



### 4.2. Wet Tropospheric Delay

To assess the accuracy of WTC, we employ the 'cross-over approach' outlined by Legeais et al. (2014). The idea of this approach finding cross-over points over the same region within a relatively short time period (10 days or below). The sea

surface height (SSH) is assumed to be on average constant within this relatively short time interval. Thus, sequential SSH observations should ideally give the same answer within the expected uncertainties. Based on this analysis method, the accuracy of different WTC retrievals can be compared. In particular, the WTC retrieval showing the smallest variability in retrieved SSH will be most accurate.

Here, we compare our WTC retrieval with two independent retrievals. Following Legeais et al. (2014), we compare our

WTC-retrieval firstly to WTC calculated only on the basis of ERA-Interim observations. Secondly, we compare to ESA's operational retrieval, which employs a neural network for WTC retrieval and, importantly, uses the MWR brightness temperatures together with altimeter-derived information about the state of the sea surface (the altimeter backscatter coefficient). This additional information allows for a better characterization of surface emissivity and is not used in our retrieval.

Figure 6 summarizes the validation results for Envisat. Comparisons against ERA-Interim WTC show overall improvements over the entire data record consistent with similar results shown in Legeais et al. (2014) for other sensors. The lower left panel of Figure 6 shows improvements nearly everywhere over the oceans with the exception of the Gulf Stream off the coast of North America, the Falkland current and the confluence areas of the Agulhas and Benguela currents. All of these areas show exceptionally high variability in SSH and might therefore pose particular challenges to the validation method

used here.

Comparing our retrievals with the operational Envisat retrievals (Figure 6, upper and lower right panels), our algorithm performs slightly inferior than the operational algorithm in most areas, and only shows some improvements in the Southern Ocean near the ice edge. Between the three satellites studied here, Envisat shows the strongest deterioration of results compared to the operational retrieval. In fact, ERS-1 and ERS-2 show consistent improvements over the operational retrieval

in the northern and southern mid-latitudes (see Figure 7). This result is surprising in particular, as our algorithm does not use any altimeter information to constrain sea surface emissivity and solely relies on the MWR observations and ERA-Interim as background information.

### 5. Conclusions

We have established a new long-term (1992-2012) dataset of TCWV and WTC over the global oceans, which is readily

available for users. It has also been submitted to the ongoing GEWEX Water Vapour Assessment (G-VAP, Schroeder et al. (2016)) for evaluation in the context of several other TCWV datasets. Our validation against GNSS shows the TCWV




dataset to be stable over time. WTC also provides promising results, although ESA's operational WTC retrieval is slightly better in particular for Envisat. In contrast to our retrieval ESA's retrieval does use additional collocated altimeter backscatter information to constrain sea surface emissivity. Our algorithm currently relies only on ERA-Interim surface wind speed. A 1D-VAR retrieval of surface wind speed, water vapour, and cloud liquid water based on combined radar altimeter

and MWR data is envisioned as a next step. This will likely have a significant positive impact on retrievals from both instruments. It would require extending the surface emissivity model as well as the 1D-VAR retrieval to include altimeter backscatter.

A number of further limitations exist: In the current implementation, ERA-Interim fields are only used once a day at 12 UTC

as background profiles. Rapid changes of atmospheric conditions and surface properties are not accounted for. Therefore, we have studied the impact of the background state on retrieval quality (see Appendix B) and conclude that this limitation has only a marginal impact on data quality, as the algorithm is only weakly dependent on the choice of the background.

Since MWR is nadir looking only, it does not provide any polarization information. Compared to other microwave sensors, its spectral range is also limited to frequencies below 37 GHz. Therefore, screening observations affected by frozen

hydrometeor scattering will not be possible. Thus, in cases of moderate to heavy frozen hydrometeor load, such as in deep convective cores, retrieval results will likely be degraded. Here we employ a screening based on the final value of the cost function, which has proven efficient in eliminating outliers. However, validation results show that the comparisons against GNSS are deteriorated for larger LWP, which would be indicative of remaining issues with precipitation screening.

Finally, an extension of the approach outlined here to include the MWR on-board the Sentinel-3 satellites and potentially

other altimeter/radiometer combinations appears highly desirable and would allow extending the current time series forward in time. Feedbacks between improved calibration efforts for single instruments and subsequent inter-calibration needs to be accounted for. Ideally, a processing chain would be set up allowing for re-processing the inter-calibration for a single instrument, if the underlying calibration for that instrument had improved.



## Appendix A – Calculating WTC

### General overview

The altimeter path delay along a path H is directly related to the real part of the refractive index of moist air n:

$$\Delta z = \int_0^H (n-1)\,dz \tag{2}$$

Expressing this it terms of refractivity N, with N in ppm being:

$$N = 10^6 (n-1) \tag{3}$$

we get:

$$\Delta z = 10^{-6} \int_0^H N\,dz \tag{4}$$

Assuming H is the satellite altitude, nadir view, Tv to be the virtual temperature, and using hydrostatic equilibrium we get:

$$\Delta z = 10^{-6} \frac{R_{AIR}}{g} \int_0^{p_{SFC}} N \frac{T_v}{p}\,dp \tag{5}$$

The refractivity N can be parameterized following references cited in (Mangum, 2009):

$$N = a_d \frac{p_d}{T} + a_w \frac{e}{T} + b_w \frac{e}{T^2}$$

$$a_d \quad : \quad 0.776890 \quad \frac{ppm \cdot K}{Pa}$$

$$a_w \quad : \quad 0.712952 \quad \frac{ppm \cdot K}{Pa}$$

$$b_w \quad : \quad 3754.63 \quad \frac{ppm \cdot K^2}{Pa} \tag{6}$$

The variable p is the total pressure and pd represent the pressure of dry air, where the total pressure p = e+pd, with e being

20    the water vapour partial pressure. With these definitions we can write the total path delay as:



$$\Delta z = 10^{-6} \frac{R_{AIR}}{g} \left[ \int_0^{p_{SFC}} a_d \frac{p_d}{T} \frac{T_v}{p} dp + \int_0^{p_{SFC}} a_w \frac{e}{T} \frac{T_v}{p} dp + \int_0^{p_{SFC}} b_w \frac{e}{T^2} \frac{T_v}{p} dp \right]$$

(7)

The first term in the brackets in Equation (7) can be split as follows:

$$\int_0^{p_{SFC}} \frac{p-e}{T} \frac{T_v}{p} dp = \int_0^{p_{SFC}} \frac{p}{T} \frac{T_v}{p} dp - \int_0^{p_{SFC}} \frac{e}{T} \frac{T_v}{p} dp$$

(8)

so that Equation (7) can be expanded to become :

$$\Delta z = \underbrace{10^{-6} \frac{R_{AIR}}{g} \int_0^{p_{SFC}} a_d \frac{p}{T} \frac{T_v}{p} dp}_{\text{Dry tropospheric delay}} + \underbrace{10^{-6} \frac{R_{AIR}}{g} \left[ \int_0^{p_{SFC}} (a_w - a_d) \frac{e}{T} \frac{T_v}{p} dp + \int_0^{p_{SFC}} b_w \frac{e}{T^2} \frac{T_v}{p} dp \right]}_{\textit{Wet} \text{ tropospheric delay}}$$

(9)

Note that $T_v / T \simeq 1$.

**Dry delay**

Integrating the dry tropospheric part of Equation (9) yields:

$$\Delta z_d = 10^{-6} \cdot \frac{R_{AIR}}{g} \cdot a_d \cdot p_{SFC}$$

(10)

The dry delay is in the order of 2.3 m for a straight vertical path through the atmosphere whereas the wet tropospheric delay is only in the order of 0.4 m at maximum.

**Wet delay**

Integrating the wet tropospheric terms in Equation (9) yields:



$$\Delta z_w = 10^{-6} \frac{R_{AIR}}{g} \left[ \int_0^{p_{SFC}} (a_w - a_d) \frac{e}{T} \frac{T_v}{p} dp + \int_0^{p_{SFC}} b_w \frac{e}{T^2} \frac{T_v}{p} dp \right]$$

(11)

The water vapour mass mixing ratio is defined as:

$$r = \frac{R_{H2O}}{R_{AIR}} \frac{e}{p}$$

(12)

Replacing e/p accordingly with r into Equation (11) yields:

$$\Delta z_w = 10^{-6} \frac{R_{H2O}}{g} \left[ (a_w - a_d) \int_0^{p_{SFC}} r\, dp + b_w \int_0^{p_{SFC}} \frac{r}{T} dp \right]$$

(13)

10  The total column water vapour (TCWV) is defined as:

$$TCWV = \frac{1}{g} \int_0^{p_{SFC}} r\, dp$$

(14)

We further define a 'water-vapour-averaged mean inverse atmospheric temperature', $T_m$:

$$T_m = \left( \int_0^{p_{SFC}} \frac{r}{T} dp \, / \, TCWV \right)^{-1}$$

(15)

With these two quantities, Equation (13) becomes:

$$\Delta z_w = 10^{-6} R_{H2O} \left( (a_w - a_d) + \frac{b_w}{T_m} \right) \cdot TCWV = \left( A + \frac{B}{T_m} \right) \cdot TCWV$$

$$A \quad : \quad 10^{-6} \cdot R_{H2O} \cdot (a_w - a_d) \quad : \quad -2.95077 \cdot 10^{-5} \quad [m/(kg/m^2)]$$

$$B \quad : \quad 10^{-6} \cdot R_{H2O} \cdot b_w \quad : \quad 1.73276 \quad [m/(K \cdot kg/m^2)]$$

(16)

The wet tropospheric delay is in the order of 0.4 m for high water vapour content.



**Appendix – B Sensitivity to a-priori and background error covariance matrix**

The observation and background error covariance matrices perform two major functions, namely:

- Firstly, the trace of the observation error covariance matrix compared to the trace of the background error covariance matrix determines the relative weight of the observations compared to the background.

- Secondly, the background error covariance matrix determines the relative weight of the individual entries in the state vector with respect to each other. Similarly, the observation error covariance matrix determines the relative weight of the single observations in the observation vector with respect to each other. These 'internal' weights are independent of any common multiplicative factor that can be extracted from the error covariance matrix.

Thus, by multiplying e.g. the background error covariance matrix by a scalar factor larger (smaller) than unity, the 1DVAR can be forced to converge farther away from (closer to) the a-priori. This property of the covariance matrices is of importance for example when a reanalysis background is exchanged for a climate background profile, the former likely being a much better representation of the actual state of the atmosphere than the latter.

**Sensitivity studies**

Subsequently, we study the impact of fixed versus variable a-priori as well as the impact of relaxation of the background error covariance matrix over the values established in earlier studies (Schröder et al., 2013), which are used as a reference. In order to understand the sensitivity of the retrievals to different choices of background state vectors (xb) and background error covariance matrices (Sb) the following series of retrieval tests was performed on a single day Envisat MWR observations (2011/12/01). The following four tests were run:

1. ERA_OLD: ERA-Interim background with existing constraints (see below) on water vapour and liquid water background error covariance.

2. ERA_NEW: ERA-Interim background, but with less tight constraints on water vapour and liquid water. Constraints were relaxed by a factor of two in water vapour and five in cloud liquid water.

3. FIXED_OLD: A fixed mid-latitude summer atmospheric background profile was used for all retrievals. The same existing constraints on error covariance matrices as in 1 were used.

4. FIXED_NEW: The same fixed background profile as in 3, but with less tight constraints on background error covariance (as in 2).

The fixed background was simulated using a mid-latitude standard atmosphere as first guess and background for all retrievals. Only surface wind speed and sea surface temperature were still used from ERA-Interim for the climate background. The FIXED scenario represents an extreme case of a climatological background profile in which the background is kept fixed regardless of location and season. This extreme case has been chosen because it allows studying the algorithm performance under a most restrictive scenario with a fixed background.





The modified background error covariance matrix for tests 2 and 4 was implemented expanding the background standard deviation for cloud liquid water from 0.2 kg/m$^2$ to 1.0 kg/m$^2$ and in addition by multiplying the lnQ sub-matrix of the background error covariance by a factor of two, where lnQ is the logarithm of the water vapour mixing ratio which is the water vapour control variable in the applied 1D-VAR retrieval scheme. For all cases we counted retrievals as valid, if, after

convergence, the final cost function value was lower than five, i.e. at maximum 2.5 times larger than the expectation value of the cost function for valid retrievals.

The results of these tests are summarized in Table 2 and Figure 8. Both FIXED retrievals show critical deficiencies. Neither the original nor the modified settings allow for a good fit using just one constant climate-like profile as background. In both FIXED cases only about 50% of the retrievals actually converge and large biases occur both at the high and low end of

TCWV. These issues can be mitigated by further increasing the lnQ sub-matrix of the background error covariance by a factor of 10 instead of two and by increasing also the number of iterations in the minimization process from a current upper limit of five to 40. However, even with these newly revised parameters the number of converged profiles remains lower than for the ERA background.

An important finding from the FIXED cases is the relative insensitivity of the retrieval to the choice of the background. As

can be seen in Figure 8, as long as the actual TCWV is less than maybe 5-10 kg/m2 away from the chosen background, the retrieval will perform quite well, especially under relaxed background error covariance conditions. This is due to the high information content of the passive microwave observations with respect to both TCWV and LWP.

We note that the use of a single global background profile is not necessarily the best choice for a climatological background. A possible compromise could consist of less stringent choices of climatological backgrounds allowing the background water

vapour and temperature profile to vary with geographical position and latitude.

Compared to the FIXED cases both the ERA-NEW and ERA-OLD case show much better results. The trade-off here lies mainly between an increased RMSE (NEW) and an increased number of profiles with large remaining Tb residuals after convergence (OLD). The ERA_NEW case allows the 1DVAR to find low cost solutions further away from the background water vapour profile. This will enhance the RMSE because we compare the retrieved TCWV to the background TCWV.

ERA_NEW in contrast provides tighter constraints on the background error covariance matrix, thus minimizes the RMSE better but at the cost of having a larger fraction of retrievals not converge as closely toward the observed brightness temperatures. For the particular case shown here 6.24 % of retrievals still show a residual deviation of simulated from observed Tb-s larger than 1 K.

A design choice for the final retrieved time series was therefore the extent to which it adheres to the prescribed background

compared to perfectly minimizing the observed brightness temperatures. We note that in an ideal world with perfect knowledge about background and observation error covariance matrix this choice could not be made and the retrieval would provide a perfect a-posteriori estimate of the true state of the atmosphere accounting for correct background and observational information. However, as is always the case the actual retrieval will have to be tuned to some degree. In particular, one wants to minimize the risk of artifacts in the background data to affect the final TCWV time series. Such




artifacts can example includes slight discontinuities in the ERA TCWV time series, at time steps where new sensors are added to the reanalysis.

With these considerations in mind we have chosen the ERA_NEW 1DVAR setup to be used as the basis for the full time series. The modified background error matrix allows for large deviations from the background profile, i.e. it gives stronger

weight to the observations. At the same time, it provides good convergence over the entire range of variability of TCWV and allows for a high number of converged profiles and therefore provides very little sensitivity to the choice of the background profile. In particular, the choice of the background profile is uncritical, as long as it represents the general conditions for the geographical region and season.

**--- Figure 8 here ---**

                                **--- Table 2 here ---**

**Acknowledgements**

This work was funded under ESA's Long Term Data Preservation Program (LTDP). Partial funding was also provided by

Vanderbilt University via a start-up grant to the first author.



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





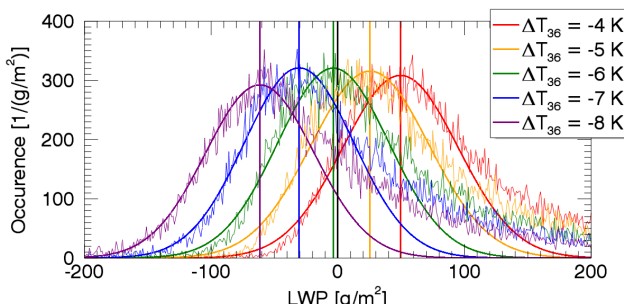

**Figure 1: Example of histograms of retrieved LWP and fitted Gaussian for different bias correction values at 36 GHz. The example is shows for MWR on Envisat, January 2001. For the example shown here, the bias correction at 23 GHz is set fixed to -3 K.**





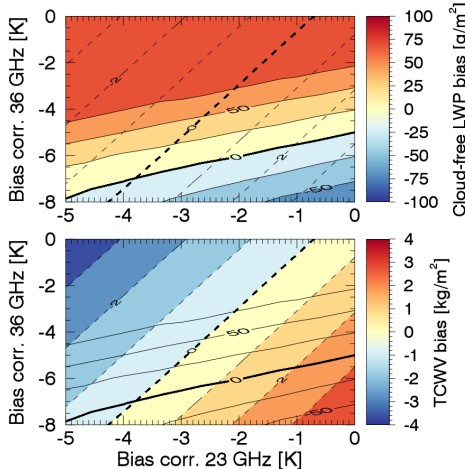

**Figure 2: The colored contour plot in the upper panel shows contours of LWP bias as function of bias correction values for 23 GHz (x-axis) and 36 GHz (y-axis). The colored contour plot in the lower panel shows biases in TCWV. In addition, labeled isolines of both TCWV-bias (dashed) and LWP-bias (solid) are overlaid in both plots.**





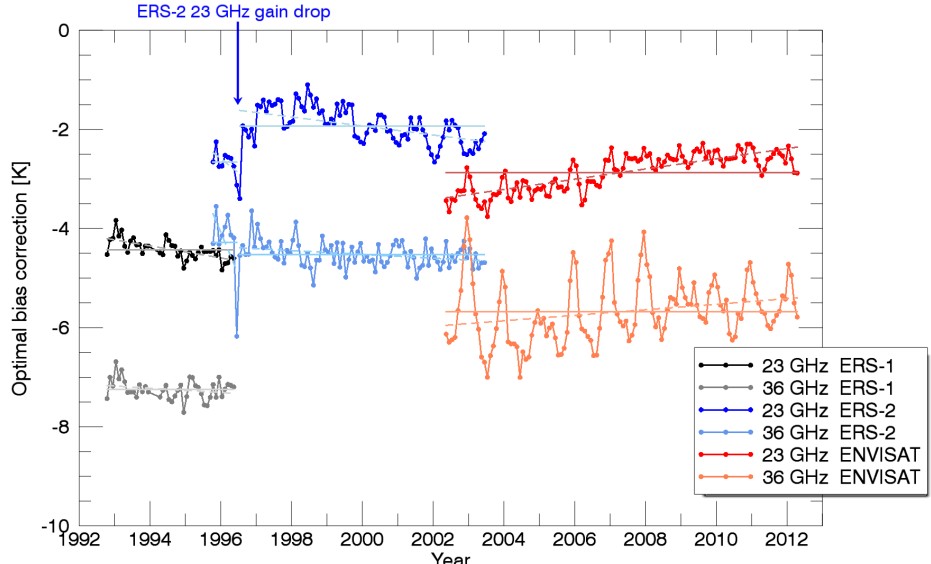

**Figure 3: Optimal bias correction values for the entire time series shown as colored lines with filled circles marking each monthly value. Also shown are the temporal mean values for each channel and instrument (straight lines in slightly lighter colors than the corresponding monthly values) and a regression line (dashed lines). For ERS-2 two separate fits were performed. One corresponding to the period before the 23 GHz gain drop and another one for the period after the gain drop. The blue arrow highlights the time the day drop happened (6/26/1996). A negative bias correction value means that the observations are warmer than the simulations, thus need to be corrected downwards.**





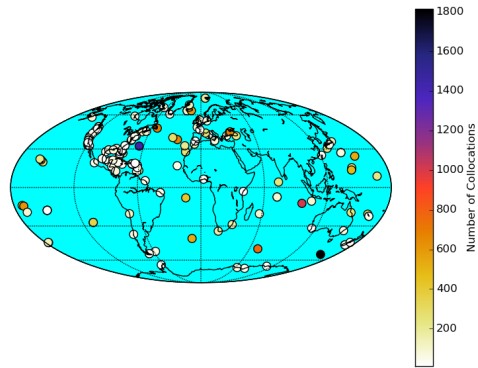

**Figure 4: Overview of location and data density of GNSS stations used for TCWV analysis.**





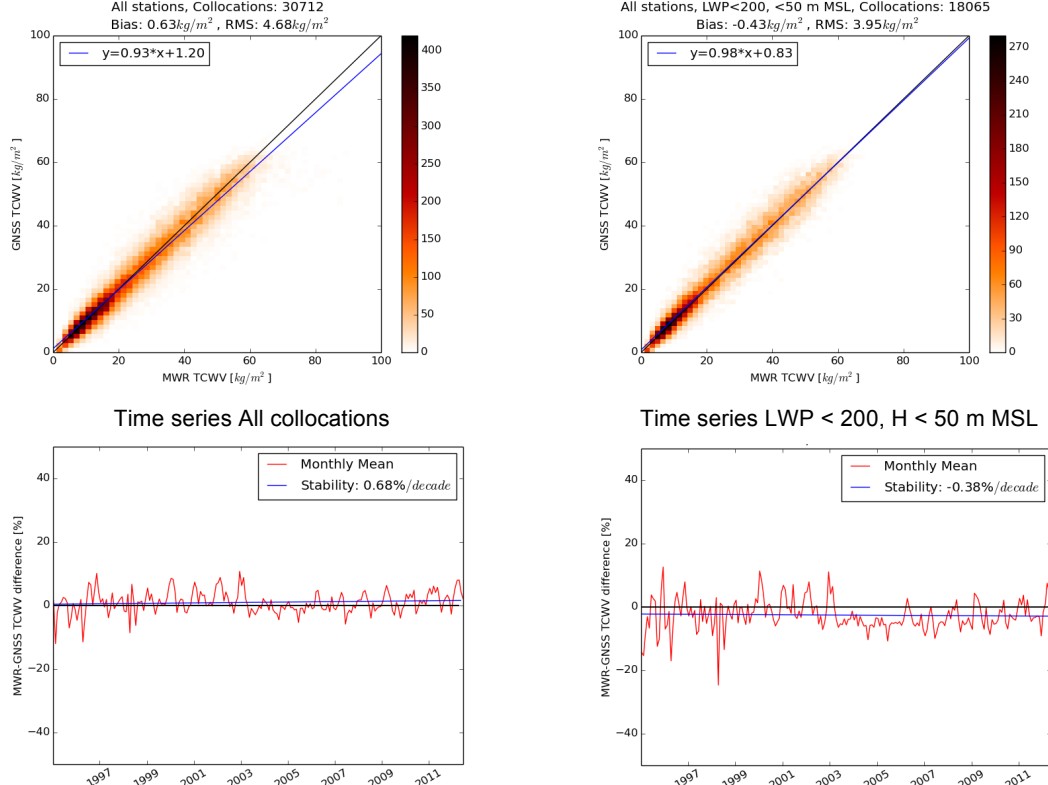

**Figure 5: Comparison of GNSS and MWR-derived TCWV. The left panels show comparisons for all collocations. The right panels show comparisons for a sub-set of collocations where (1) MWR-retrieved LWP was smaller than 200 g/m$^2$ and (2) the GNSS station height was below 50 m above MSL. The top panels show data density plots. The bottom panels show time series of monthly-mean differences between MWR and GNSS for the entire dataset. The blue lines give linear fits to the data.**





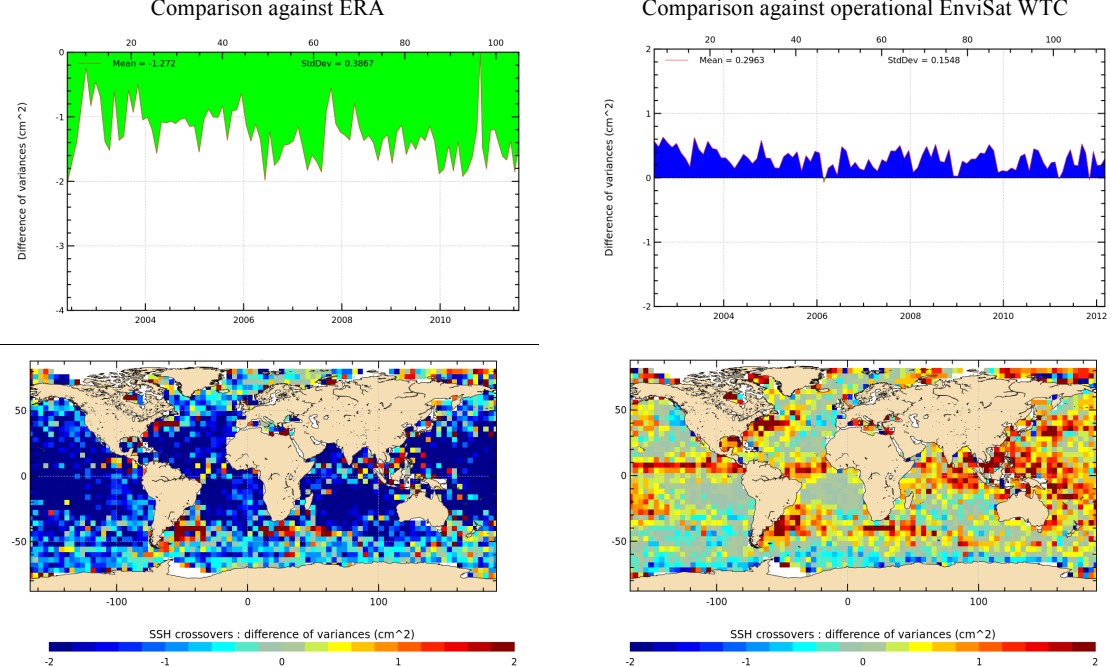

**Figure 6: Validation statistics for WTC sea surface height variability. Data shown here are for Envisat only. The left panels show comparisons against WTC derived purely from ERA-Interim. The right panels show comparisons against the operational WTC retrieval from EnviSat. The upper plots shows globally averaged monthly statistics. The lower plots show long-term spatial statistics. Values smaller than zero indicate an improvement. Values larger than zero indicate a deterioration of the results relative to the comparison dataset.**





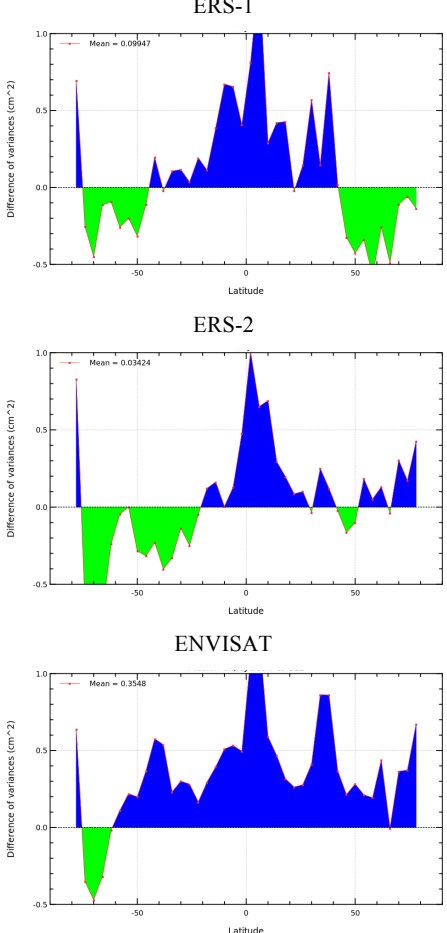

**Figure 7: Zonally averaged patterns of improvement (green) and deterioration (blue) of our retrieval over the operational ESA retrieval.**





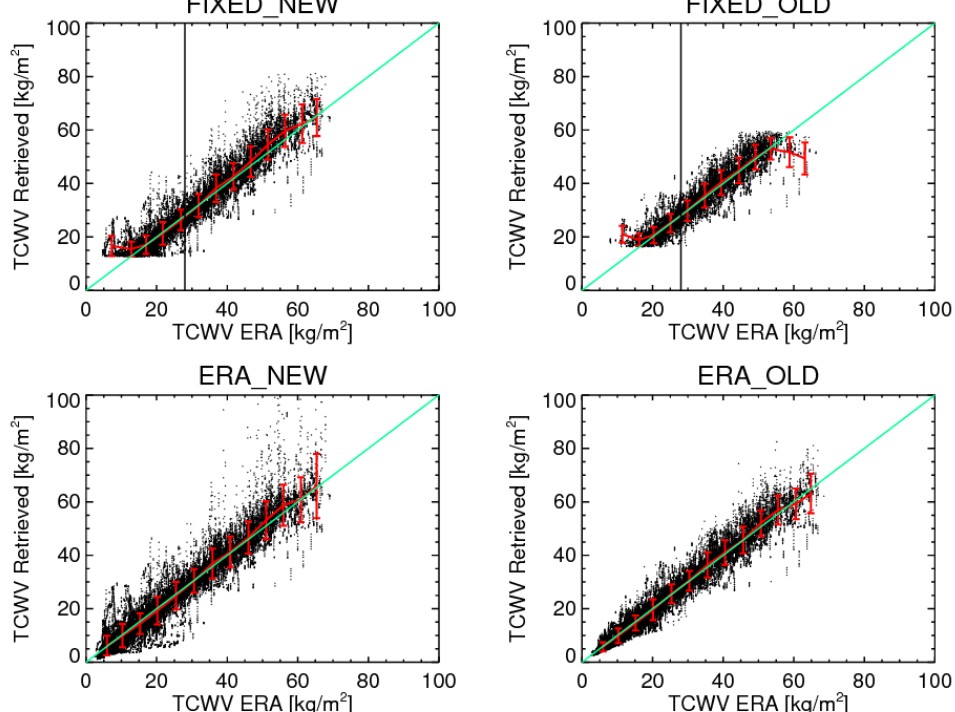

**Figure 8:** Scatterplots of retrievals obtained with the four different 1DVAR configurations described in Section 2.3.1. In all four panels the retrieved TCWV is plotted against ERA-Interim TCWV. The green line is the 1-1 line. The red error bars show the mean and standard deviation in bins of 5 kg/m2. The vertical line in the upper two plots shows the TCWV of the fixed climate background used for these plots (mid-latitude summer atmosphere). Total number of retrievals was 35,584. Reported values are for valid retrievals with cost function lower than 5. Corresponding statistics are listed in Table 2.



**Table 1: Mean optimal bias correction values and regression slopes for the all instruments and time periods. The values given here correspond to the straight lines and dashed lines in Figure 3. A negative bias correction value means that the observations are warmer than the simulations, thus need to be corrected downwards.**

| Instrument | Period | Mean | | Regression Slope(*) | | Regression Offset(*) | |
|---|---|---|---|---|---|---|---|
| | | 23 GHz | 36 GHz | 23 GHz | 36 GHz | 23 GHz | 36 GHz |
| | | [K] | [K] | [K/yr] | [K/yr] | [K] | [K] |
| ERS-1 | 10/1992 – 06/1996 | -4.42 | -7.25 | -0.12 | -0.04 | -3.86 | -7.05 |
| ERS-2 | 10/1995 – 06/1996 | -2.66 | -4.28 | -0.57 | -1.72 | +0.83 | +6.24 |
| ERS-2 | 07/1996 – 06/2003 | -1.93 | -4.52 | -0.09 | -0.04 | -1.02 | -4.14 |
| Envisat | 05/2002 – 04/2012 | -2.87 | -5.68 | +0.10 | +0.06 | -4.65 | -6.65 |

**(*) The regression bias is calculated using bias_corr(t) = slope *t + offset, where t is the decimal year since 1990. For example, July 2, 1991 is day 183 in the year 1991 and therefore corresponds to a value of t=1.5.**



**Table 2: Retrieval statistics for the four different 1DVAR configurations described in Section 2.3.1. Corresponding scatterplots are shown in Figure 8. Total number of retrievals was 35,584. Reported bias and RMSE values are for valid only for the fraction of retrievals with cost function lower than 5.**

| Experiment | Bias with respect to ERA-Interim | RMSE with respect to ERA-Interim | Percent retrieved | Mean Tb residual after retrieval | Percent with Tb residual larger than 1 K. |
|---|---|---|---|---|---|
| | [kg/m2] | [kg/m2] | [%] | [K] | [%] |
| FIXED_NEW | 0.88 | 4.80 | 68.5 | 0.23 | 0.02 |
| FIXED_OLD | 0.46 | 3.99 | 45.7 | 0.42 | 1.52 |
| ERA_NEW | -0.01 | 5.06 | 97.9 | 0.07 | 0.91 |
| ERA_OLD | 0.00 | 3.53 | 87.2 | 0.41 | 6.24 |

