# Peer review of "An intercalibrated dataset of Total Column Water Vapour and Wet Tropospheric Correction based on MWR on board ERS-1, ERS-2 and Envisat"

_Atmospheric Measurement Techniques, 2016_

## Referee Comment (RC1) · Anonymous Referee #1 · 10 Nov 2016

This short paper introduces a retrieval of total column water vapour (TCW), liquid water path (LWP) and wet tropospheric correction (WTC) for three microwave radiometers. As part of this study an intercalibration of this datasets is introduced. A comparison of TCW versus TCW measured with ground based GPS is presented.

The subject matter of this paper is appropriate for AMT as no science is really discussed. The manuscript is overall well written, however some important details regarding the retrieval and intercalibration needs to be expanded (see below). The comments below should be addressed before publication.

Section 2.1 needs to include a more detailed satellite description: launch dates, termination dates, type of orbits, etc. It also needs to include more details on the microwave

radiometers (MWR): location of the channels, width, footprint size, are the channels exactly the same for the three instruments? Error characterization for the channels (expected error in brightness temperature), lat lon coverage in an average day, numbers of measurements per day, etc.

Section 2.3.1 needs to be expanded to include details on the forward model, spectroscopy used, do you use the Liebe model or something similar? mention the gases absorbing in this region? mention why you need the surface air temperature, surface specific humidity and the wind speed? which model do you use for sea surface roughness? Are you sure that the LWP from ERA-interim is accurate? How many iterations are needed for convergence?, which apriori errors are assumed in Sb, is So assumed to be diagonal? Which cost function value is used as convergence criteria?

Section 3.1: Please explain how you select the 4% of the observations, could there be a sampling bias? Are they distributed through out the globe?

Section 4. Please include a comparison against ERA-Interim vs the GNSS stations as well as versus the ERS1, ERS2 and ENVISAT data. The purpose of this comparison is to see if the new dataset posses more information than ERA-Interim (i.e., hopefully the comparison of the new datasets vs the GNSS stations is going to be better than the ERA-Interim / GNSS comparison)

Further, LWP has not been validated. Comments on its usefulness are needed.

Appendix B

The sensitivity study needs to be expanded further. No information is given on the impact of the surface wind speed and sea surface temperature.

The statement "as long as the actual TCWV is less than maybe 5-10..." needs to be proved (change the background by 5 -10 km/m2 randomly through-out the vertical profile and do the scatter plots).

What is the impact of the temperature being off by 3, 5, 10K.

Again, prove the statement: the choice of background is uncritical as long as it represents the general conditions for the geographical region and season.

Specific Comments

P2 L2: Define ERS

P3 L3: Define GNSS (it was defined in the abstract but it needs to be defined again in the text)

P3 L12: Define CLS (currently, only defined in the affiliations)

P3 L13 please explain the anomaly and if know, the cause of the anomaly.

P3 L22: Define GNSS in P3 L3 not here.

P4 L2: Should this section be called TCWV and LWP?

P4 L8: Could you please explain if this retrieval is exactly the same as optimal estimation (it looks like it) or if there is any difference could you mention those.

P4 L10: Please give some basic details about ERA-Interim or at least provide a reference.

P4 L13: Mention that the channel at 23 is close a line emission center and that the channel at 36 is a window channel.

P5 L11: Appendix B seems to indicate that the retrieval is based on $\ln(q)$ which will make it impossible to get negative values for TCW please explain why TCWV>0 is possible.

P5L12: Why such a big value of a cost function shouldn't this be close to 1?

P8 L13: Is this 3 a footnote? It needs to be a superscript. Also, which figure in this link is related to your statement. Please clarify further.

P8L29: Delete extra dot.

P9L4: Define MSL.

P11L14: delete frequencies below 37 GHz and change to limited to two frequencies 23 and 36 GHz.

P15L17: xb needs to be bold because it is a vector and the b needs to be a subscript.

P15L18: Sb needs to be bold because it is a matrix and the b needs to be a subscript.

Figure1 caption: Envisat was launch in 2002 so the date most be wrong or this is a different satellite, please check.

Figure 2: There is redundant information on both panels please delete one. Also specify for which satellite/month this is.

Figure 5 top add number of co-locations to the color bar.

Figure 5 bottom right: there is a clear bias in the time series not shown in the scatter plot above please fix.

---

## Referee Comment (RC2) · Anonymous Referee #2 · 25 Nov 2016

The paper presents a new water-vapour data record for climate, compiled from a series of different microwave missions. The manuscript is found well structured, clearly presenting in sequence the background information on the objectives and instrumentation, the measurements and their calibration, the retrieval methodology and in the end some validation results. The manuscript will be an important documentation of the dataset agregating results from a series of different missions to compile a climate data record of water-vapour. It is recommended to consider the comments below before publication.

A General comment: the methodology deployed in this paper seems to clearly indicate that anchors are defined on a monthly basis on ERA-Interim to perform the bias cor-

[Figure]

rection of the microwave measurements. In turn, the TCWV retrieval is expected to be subsequently anchored on average on the model itself. This deserves some more discussions in the paper and clear statements as to what added-value this TCWV dataset is bringing in the scope of climate studies, wrt trend assessment of average quantities for instance.

Specific comments (Px.Ly stands for page x and line y)

Abstract L30-L32: not clear which products are referred to when stating on 'superiority' (L30) and 'improvements' (L31). is it compared to previous versions of the same processor, to numerical model forecasts/analyses, to other intruments' WTC products, to other institutes' MWR WTC products ? I suggest making specific references here to highlight more to what the proposed dataset is adding value.

p3.L3: typo "it's" –> "its"

p3.L14: suggest adding a reference were the bias in v2.1 is characterised.

p4.L3, editorial: first sentence is confusing

p4.L16-17, editorial: unnecessary repetition of the same info

p4.L24: were the uncertainties on the fixed SV parameters taken into account in the observation error matrix?

p5.L2: Rodgers 2000 does not explain specifically how the Sa and So matrices were established for this particular retrievals. It is essential information to understand the new product. In particular also if that differs from the retrieval methodologies in other products. It is said in introduction that So includes forward modelling error and instrument noise. How was this estimated? Similarly, how was Sa determined?

p5.L11: does the retrieval methodology produce negative TCWV ? That reads odd, clarification of why/what is meant here may be required at this stage.

p6.L25: Is this empirical bias correction assessment performed on cloud-free pixels

only ? This is what I would guess from the statements made in the bullet point above. If some cloud screening was applied it should be made explicitly clear here and described. Figure 1: The fitted Gaussian on the main mode is presumably covering the cloud-free scenes. The Authors explain that the negative values result from the random instrument noise. At the same time the positive values outside this Gaussian fit are associated to cloudy scenes. However, the negative values go as low as 100g/m2, which is of the same magnitude (in absolute terms) as the LWP in cloudy cases. This suggests that the effect of the instument noise has a very strong and direct impact on the precision of the retrieved quantities including in cloudy scenes. Can the Authors comment on this and what limitations this has in view of the climate application sought here? Possibly reflect some of this in the manuscript?

p6.L28: The reader would benefit from a brief explanation about the rationale and purpose of the 4% subsampling

P8.L29: typo, double '. '

Conclusions: it is not clear why the WTC record is still proposed by the Authors whilea, according to their assessment, this dataset is showing less skills than the operational one established by ESA. It is strongly recommended to elaborate and highlight more specifically the potential advantages of this dataset or the necessity to provide this independent record, for clarity to the reader.

---

## Author Comment (AC1) · 28 Jan 2017

**amt-2016-304**

**An intercalibrated dataset of Total Column Water Vapour and Wet Tropospheric Correction based on MWR on board ERS-1, ERS-2 and Envisat**

**Ralf Bennartz et al.**

**Response to Reviewer 1**

**We thank the reviewer for his constructive comments. Below please find a detailed response on all reviewer comments.**

Section 2.1 needs to include a more detailed satellite description: launch dates, termination dates, type of orbits, etc. It also needs to include more details on the microwave radiometers (MWR): location of the channels, width, footprint size, are the channels exactly the same for the three instruments?

**We have added this information now at the beginning of Section 2.1 and have also added a new Table 1 that summarizes key information about the instruments.**

Error characterization for the channels (expected error in brightness temperature),

**We have added this information in Appendix B and toward the end of Section 3.1**

lat lon coverage in an average day, numbers of measurements per day, etc.

**We have added this information now at the beginning of Section 2.1 and have also added a new Table 1 that summarizes key information about the instrument.**

Section 2.3.1 needs to be expanded to include details on the forward model, spectroscopy used, do you use the Liebe model or something similar? mention the gases absorbing in this region? mention why you need the surface air temperature, surface specific humidity and the wind speed? which model do you use for sea surface roughness?

**We have added a statement and two references. We are using the RTTOV forward modelling package, which is being used routinely by a huge number of operational and research users. All the information requested by the referee can be found in the RTTOV documentation.**

Are you sure that the LWP from ERA-interim is accurate?

**We are not using the ERA-Interim LWP at all. We only use the profile of LWP to distribute liquid vertically. (The only alternative would be to just select a height range in which to put liquid clouds, which to us appears even worse).**

**Once the vertical profile of liquid is established, we initialize the retrieval with a first guess LWP of 30 g/m$^2$ and then let the retrieval take over. As pointed out in the sensitivity studies, we are virtually independent of the first guess (as we should be).**

How many iterations are needed for convergence?

**On average the algorithm converges after less than five iterations.**

Which apriori errors are assumed in Sb, is So assumed to be diagonal?

**This is discussed in Appendix B, where we refer to the appropriate reference. We have now included a statement in the main text referring to that.**

Which cost function value is used as convergence criteria?

**At level 2 we do not screen according to cost function. When aggregating up to gridded data, we use a cost function value of five. This is addressed in Section 2.4.**

Section 3.1: Please explain how you select the 4% of the observations, could there be a sampling bias? Are they distributed through out the globe?

**This was just a compromise between having enough data for meaningful statistics and computational efficiency. We have added a statement to that extent.**

Section 4. Please include a comparison against ERA-Interim vs the GNSS stations as well as versus the ERS1, ERS2 and ENVISAT data. The purpose of this comparison is to see if the new dataset posses more information than ERA-Interim (i.e., hopefully the comparison of the new datasets vs the GNSS stations is going to be better than the ERA-Interim / GNSS comparison)

**This is an excellent suggestion and we have included the requested information in the new Table 2. The results indeed show an improvement in terms of a reduced RMS-Error.**

Further, LWP has not been validated. Comments on its usefulness are needed.

**We have added some discussion on LWP and expected uncertainties in Section 3.1, where we discuss the cloud-free LWP histograms. Beyond this, we do not see any way of 'validating' LWP as there is no reference available for such a validation. Comparisons with visible/near infrared estimates of LWP could be performed but have their own caveats and are typically only meaningful, if the two observations are on the sample platform, such as AMSR and MODIS. For the three satellites discussed here, that option does not exit. For more details on general LWP comparisons see e.g.:**

- **Seethala, C., and Horváth, Á.: Global assessment of AMSR-E and MODIS cloud liquid water path retrievals in warm oceanic clouds, Journal of Geophysical Research, 115, 10.1029/2009jd012662, 2010.**
- **Greenwald, T. J.: A 2 year comparison of AMSR-E and MODIS cloud liquid water path observations, Geophys Res Lett, 36, 10.1029/2009gl040394, 2009.**
- **Borg, L. A., and Bennartz, R.: Vertical structure of stratiform marine boundary layer clouds and its impact on cloud albedo, Geophys Res Lett, 34, DOI: 10.1029/2006GL028713, 2007.**

Appendix B The sensitivity study needs to be expanded further. No information is given on the impact of the surface wind speed and sea surface temperature.

**It is unclear to us, what such a sensitivity study would achieve since those parameters are fixed parameters that are not retrieved. In general, there are various studies already out there that address the sensitivity of MW retrievals w.r.t. wind speed and sea surface temperature. For the two frequencies under consideration, see e.g.:**

> **Ellison, W. J., English, S. J., Lamkaouchi, K., Balana, A., Obligis, E., Deblonde, G., Hewison, T. J., Bauer, P., Kelly, G., and Eymard, L.: A comparison of ocean emissivity models using the Advanced Microwave Sounding Unit, the Special Sensor Microwave Imager, the TRMM Microwave Imager, and airborne radiometer observations, J Geophys Res-Atmos, 108, 10.1029/2002jd003213, 2003.**

The statement "as long as the actual TCWV is less than maybe 5-10..." needs to be proved (change the background by 5 -10 km/m2 randomly through-out the vertical profile and do the scatter plots).

**Figure 8 and the corresponding table actually show exactly this. For a FIXED background of 28 kg/m2 all retrievals with an actual TCWV between about 20 and 40 kg/m2 do converge. We have rephrased the sentence slightly to make clearer what we actually meant.**

What is the impact of the temperature being off by 3, 5, 10K.

**If the reviewer asks for the general sensitivity of retrieved parameters to changes in Tbs, those can be read off Figure 2 and are discussed also toward the end of Section 3.1. We have added some more discussion there, that also justifies our choice for the magnitude of the observation error covariance matrix.**

**Good rule of thumb:**

$$dTCWV/dTb23 \sim 1 \text{ kg/}((m^2K))$$
$$dLWP/dTB36 \sim 25 \text{ (g/}(m^2K))$$

Again, prove the statement: the choice of background is uncritical as long as it represents the general conditions for the geographical region and season.

**See our discussion of the your earlier comment: Figure 8 and the corresponding table actually show exactly this. For a FIXED background of 28 kg/m2 all retrievals with an actual TCWV between about 20 and 40 kg/m2 do converge. We have rephrased the sentence slightly to make clearer what we actually meant.**

Specific Comments

P2 L2: Define ERS

**Fixed**

P3 L3: Define GNSS (it was defined in the abstract but it needs to be defined again in the text)

**Fixed**

P3 L12: Define CLS (currently, only defined in the affiliations)

**Fixed**

P3 L13 please explain the anomaly and if know, the cause of the anomaly.

**Fixed.**

P3 L22: Define GNSS in P3 L3 not here.

**Fixed**

P4 L2: Should this section be called TCWV and LWP?

**Agreed. Fixed.**

P4 L8: Could you please explain if this retrieval is exactly the same as optimal estimation (it looks like it) or if there is any difference could you mention those.

**Yes. We added a sentence to that extent.**

P4 L10: Please give some basic details about ERA-Interim or at least provide a reference.

**We added the reference recommended by ECMWF Dee et al. (2011).**

P4 L13: Mention that the channel at 23 is close a line emission center and that the channel at 36 is a window channel.

**We added this information.**

P5 L11: Appendix B seems to indicate that the retrieval is based on ln(q) which will make it impossible to get negative values for TCW please explain why TCWV>0 is possible.

**The retrieval only returns negative values (-999) only to indicate the retrieval has failed. Any converged retrieval will be positive as stated by the reviewer. We have rephrased this sentence to make this clearer.**

P5L12: Why such a big value of a cost function shouldn't this be close to 1?

**No, the expectation value of the cost function equals the number of degrees of freedom in the dataset. In our case the number of degrees of freedom is two. For an idealized retrieval 66 % of the retrieved cost function values should be below the number of degrees of freedom in the dataset. By allowing the cost function to go up to 5, we exclude 2.5-sigma outliers.**

P8 L13: Is this 3 a footnote? It needs to be a superscript. Also, which figure in this link is related to your statement. Please clarify further.

**Yes. Footnote… Word formatting issue…We have now addressed the figure selection on the website better in the footnote.**

P8L29: Delete extra dot.

**Fixed**

P9L4: Define MSL.

**Fixed**

P11L14: delete frequencies below 37 GHz and change to limited to two frequencies 23 and 36 GHz.

**Fixed**

P15L17: xb needs to be bold because it is a vector and the b needs to be a subscript.

**Fixed**

P15L18: Sb needs to be bold because it is a matrix and the b needs to be a subscript.

**Fixed**

Figure1 caption: Envisat was launch in 2002 so the date most be wrong or this is a different satellite, please check.

**Fixed (It was January 2005)**

Figure 2: There is redundant information on both panels please delete one. Also specify for which satellite/month this is.

**We have added the satellite/month. We are not sure what the referee is referring to in terms of "redundant information". The two panels are complementary. Only the isolines appear in both panels to provide visual orientation to the reader where the 'zero-bias' isolines intercept each other.**

Figure 5 top add number of co-locations to the color bar.

**Fixed**

Figure 5 bottom right: there is a clear bias in the time series not shown in the scatter plot above please fix.

**We believe this is correct, as the y-axis of the lower plot is in relative units. We have added the following text in the revised manuscript:**

> The lower panels in Figure 5 provide time series of average relative differences between MWR and GNSS. These were calculated by first computing the relative difference for each individual observation and then averaging these individual relative differences. Note that for the same absolute difference relative differences are larger if the absolute value is smaller, e.g. for an absolute value of 20 kg/m$^2$, an absolute difference of 2 kg/m$^2$ would correspond to a 10 % relative difference, whereas for an absolute difference value of 60 kg/m$^2$ that same absolute difference would only correspond to a 3.3 % relative error. Therefore relative biases can be different from absolute biases as can be observed in Figure 5. We provide the stability here in relative terms because the World Meteorological Organization's (WMO) Global Climate Observing System (GCOS) requirements for temporal stability of TCWV are formulated in relative term as well (GCOS-107, 2006).

---

## Author Comment (AC2) · 28 Jan 2017

**amt-2016-304**

**An intercalibrated dataset of Total Column Water Vapour and Wet Tropospheric Correction based on MWR on board ERS-1, ERS-2 and Envisat**

**Ralf Bennartz et al.**

**Response to Reviewer 2**

**We thank the reviewer for his constructive comments. Below please find a detailed response on all reviewer comments.**

A General comment: the methodology deployed in this paper seems to clearly indicate that anchors are defined on a monthly basis on ERA-Interim to perform the bias correction of the microwave measurements. In turn, the TCWV retrieval is expected to be subsequently anchored on average on the model itself. This deserves some more discussions in the paper and clear statements as to what added-value this TCWV dataset is bringing in the scope of climate studies, wrt trend assessment of average quantities for instance.

**The main advantage of our dataset over the operational ESA dataset is that it actually is inter-calibrated. There has not been any attempt so far to inter-calibrate the three instruments and create a homogeneous time series. We have added some discussion of the issue in the conclusions now that hopefully brings this important point across better now.**

Abstract L30-L32: not clear which products are referred to when stating on 'superiority' (L30) and 'improvements' (L31). is it compared to previous versions of the same processor, to numerical model forecasts/analyses, to other instruments' WTC products, to other institutes' MWR WTC products ? I suggest making specific references here to highlight more to what the proposed dataset is adding value.

**We have made these statements more specific now by adding which models and algorithms we compared to.**

p3.L3: typo "it's" –> "its"

**Fixed**

p3.L14: suggest adding a reference were the bias in v2.1 is characterised.

**Fixed. We have added a reference to a technical report that specifies this issue.**

p4.L3, editorial: first sentence is confusing

**We have split this sentence in two now.**

p4.L16-17, editorial: unnecessary repetition of the same info

**FIXED. We removed the duplicate sentence.**

p4.L24: were the uncertainties on the fixed SV parameters taken into account in the observation error matrix?

**We have added discussion the values of the observation error covariance matrix now. $S_o$ is a diagonal matrix with errors of 1 K in both channels. See also our answer to the question about LWP histograms further down, which justifies this choice.**

p5.L2: Rodgers 2000 does not explain specifically how the Sa and So matrices were established for this particular retrievals. It is essential information to understand the new product. In particular also if that differs from the retrieval methodologies in other products. It is said in introduction that So includes forward modelling error and instrument noise. How was this estimated? Similarly, how was Sa determined?

**We have now disentangled the general reference to Rodgers from the specifics of the error covariance matrices, which are discussed in Appendix B, where we refer to the appropriate reference. We have now included a statement in the main text referring to that.**

p5.L11: does the retrieval methodology produce negative TCWV ? That reads odd, clarification of why/what is meant here may be required at this stage.

**The retrieval only returns negative values (-999) only to indicate the retrieval has failed. Any converged retrieval will be positive as stated by the reviewer. We have rephrased this sentence to make this clearer.**

p6.L25: Is this empirical bias correction assessment performed on cloud-free pixels only ? This is what I would guess from the statements made in the bullet point above. If some cloud screening was applied it should be made explicitly clear here and described. Figure

1: The fitted Gaussian on the main mode is presumably covering the cloud-free scenes. The Authors explain that the negative values result from the random instrument noise. At the same time the positive values outside this Gaussian fit are associated to cloudy scenes. However, the negative values go as low as 100g/m2, which is of the same magnitude (in absolute terms) as the LWP in cloudy cases. This suggests that the effect of the instrument noise has a very strong and direct impact on the precision of the retrieved quantities including in cloudy scenes. Can the Authors comment on this and what limitations this has in view of the climate application sought here? Possibly reflect some of this in the manuscript?

**The reviewer's notion is correct. Typically, the observed standard deviation around zero for cloud-free scenes is about 30 g/m$^2$ for conically scanning microwave instruments such as SSM/I or AMSR ((Greenwald, 2009;Bennartz et al., 2010)). We reference this number in Section 3.1. In our case the width of the histograms shown in Figure 1 is about 41 g/m2, so that a negative value of, say, -86 g/m$^2$ would correspond to an about 2-sigma outlier. The above standard deviation, together with the sensitivity of the brightness temperatures to changes in LWP (Jacobians) can also yield a rough estimate on the effective noise in the retrieval. From Figure 2 ( isolines) we get:**

**dLWP/dTb23 ~ 14 (g/(m$^2$K))**

**dLWP/dTB36 ~ 25 (g/(m$^2$K))**

**Thus, a noise in Tb36 of about 41/25 = 1.7 K would yield a histogram like this, givem T23 is noise-free. Assuming both channels to have the same noise and performing error propagation on the total differential dLWP, one ends up with an estimate for the uncertainty of both T23 and T36 of ~ 1.01 K.**

**We have now briefly added a discussion of this issue in the paper. However, since LWP is not our main focus, and similar issues have been discussed already earlier, we limit this discussion to just a few sentences plus references.**

p6.L28: The reader would benefit from a brief explanation about the rationale and purpose of the 4% subsampling

**This was just a compromise between having enough data for meaningful statistics and computational efficiency. We have added a statement to that extent.**

P8.L29: typo, double '. '

**FIXED.**

Conclusions: it is not clear why the WTC record is still proposed by the Authors while, according to their assessment, this dataset is showing less skills than the operational one established by ESA. It is strongly recommended to elaborate and highlight more specifically the potential advantages of this dataset or the necessity to provide this independent record, for clarity to the reader.

**We have added several paragraphs now that hopefully explain better what the advantages and disadvantages of the new dataset are. In brief: Our dataset is inter-calibrated whereas the original ESA dataset is not. That, in our view, precludes any use of the ESA dataset for climate studies.**